# OpenReview forum: "Distilling the Knowledge in Data Pruning"
_ICLR.cc/2025/Conference — Submitted to ICLR 2025_

### Official Review · Reviewer_jiaU · 2024-10-18

**Soundness:** 2
**Presentation:** 4
**Contribution:** 2
**Rating:** 5
**Confidence:** 4

**Summary:**

This paper proposes to apply knowledge distillation techniques when a model is trained on a pruned dataset. The authors provide theoretical analysis stating that error distilling from a teacher model trained with full data will be smaller than that from a teacher trained with pruned data. Experiments on CIFAR, SVHN, and ImageNet demonstrate that applying distillation can largely enhance the performance.

**Strengths:**

1. The proposed solution is simple but effective. Merely applying dataset distillation can result in a lot of benefit on performance.
2. The writing is logical and clear. I am able to follow the method descriptions, experiments, and theoretical validation.

**Weaknesses:**

1. Although the solution is indeed simple yet effective, I do not find it surprising. After all, it is a well-known trick to apply knowledge distillation to enhance the performance, especially in the industrial community when there are insufficient data. In the academic community, there are indeed works putting forward similar insights, like [a], which also focuses on data pruning. The difference is only on sample-wise pruning in this paper and patch-wise pruning in [a].
2. It seems that the experiments are not closely aligned with the theoretical analysis. Theorem 1 would like to convey that, error distilling from a teacher model trained with full data will be smaller than that from a teacher trained with pruned data. Therefore, I expect the experiments would try to validate this point by changing $f_t$. However, almost all the experiments currently are conducted with respect to $f$.
3. Following 2, according to the proof of Theorem in the appendix, the error would monotonically decrease with the increasing of $f_t$, the data amount used for training teacher models. It seems that it cannot support the experimental finding that using a teacher model with limited capacity is better. I understand that $f_t$ indicates the data portion, which may be different from the perspective of model architecture used in experiments for various capacity. Anyway, more experimental validation with respect to $f_t$ is necessary here to support Theorem 1.

[a] On the Diversity and Realism of Distilled Dataset: An Efficient Dataset Distillation Paradigm, Sun et al., CVPR 2024.

**Questions:**

See weaknesses.

---

> ### Author Response · Authors · 2024-11-18
> **Regarding paper's novelty**
>
> Thank you for the valuable and constructive comments.
>
> As the reviewer noted, knowledge distillation (KD) is a well-known technique for enhancing a model’s performance. However, to the best of our knowledge, this work is the first to investigate the application of KD in the context of dataset pruning where the teacher is trained on the entire dataset, and the student is trained on a smaller pruned subset (we highlight the paper's scenario in figure attached in the link below).
>
> * As demonstrated in the paper, this setup reveals several novel and valuable findings, including:
>      (1) Simple random pruning outperforms more sophisticated pruning algorithms when KD is applied.
>      (2) Distilling knowledge to a student trained on pruned data exacerbates the capacity gap problem.
>      (3) The selection of the KD weight, which balances the cross-entropy loss and the KD loss, significantly impacts performance in the specific scenario of dataset pruning.
>
>
>
> * Please note that reference [a] addresses *dataset distillation*, which focuses on generating *synthetic data* to represent the original dataset (specifically, the work in [a] aims to enhance the realism and diversity of the synthetic samples generated). In contrast, dataset pruning seeks to select the most informative samples from the original dataset. This distinction is also highlighted in our related works section (lines 152–161).
>
>
> **References**
>
> [a] On the Diversity and Realism of Distilled Dataset: An Efficient Dataset Distillation Paradigm, Sun et al., CVPR 2024.
>
> [Figure_scheme](https://i.imgur.com/WcUv4Xw.png)

---

> ### Author Response · Authors · 2024-11-19
> **Regarding the question on Theorem 1**
>
> Using the same setting from [1] in Theorem 1, we assume that knowledge distillation (specifically self-distillation) improves the performance compared to vanilla training (where no teacher is used to guide the student model), as shown by [1]. Consequently, Theorem 1 states that training a student on a pruned dataset with a teacher trained on the entire dataset (i.e., ft​=1)  results in a lower error bias compared to training directly on the pruned dataset without knowledge distillation (the regular data-pruning strategy). This conclusion follows from the inequality:
>
> $$E[e(\alpha, f, f_t)]^2 \leq E[e(\alpha, f, f)]^2 \leq E[e(0, f)]^2$$
>
> where the right-most term indicates vanilla training (without knowledge distillation)
> In summary, we already know that self-distillation improves results compared to vanilla training [1]. Our theoretical analysis further demonstrates that employing a teacher trained on a larger dataset (e.g., the entire dataset) leads to additional performance gains.
> We will definitely add this clarification to the paper to better explain the motivation.
>
>
> Following the reviewer’s suggestion, we are conducting experiments to evaluate the effect of $f_t$
>  (the data fraction used to train the teacher) on the student's accuracy. We plan to share the results of this analysis soon.
>
>
> **References**
>
> [1] Rudrajit Das and Sujay Sanghavi, Understanding Self-Distillation in the Presence of Label Noise, ICML 2023

---

> > ### Author Response · Authors · 2024-11-21
> > **Additional experiments**
> >
> > Following the reviewer’s comments, we have run the suggested experiments exploring the impact of f_t on the student’s accuracy. The results highlight two key findings: (1) increasing f_t consistently enhances accuracy beyond SD; (2) in every scenario, SD surpasses standard training without KD. These observations align with the theoretical insights discussed in the theoretical section.
> >
> > We have incorporated these experiments into the main paper (see section 4.4).
> > Please note that we have uploaded a revised version of the paper, with the added text highlighted in blue.
> >
> > You can also find the results of the additional experiments in the attached photo link.
> >
> > [Add_experiments_impact_of_ft](https://i.imgur.com/QhBf9lh.png)
> >
> > &nbsp;
> > &nbsp;
> >
> > Please kindly let us know if you have any follow-up questions or areas needing further clarification.

---

> > > ### Author Response · Authors · 2024-11-25
> > >
> > > Dear Reviewer, thank you once again for your time and valuable feedback.
> > > We hope our responses have addressed your concerns. Based on your suggestions, we have added an experimental section (Section 4.4 and Figure 7) and expanded the discussion in the main paper. The revised version has been uploaded, with the new additions highlighted in blue.
> > >
> > >
> > > We would greatly appreciate it if you could revisit your evaluation and consider raising your score.

---

> > > > ### Author Response · Authors · 2024-11-28
> > > >
> > > > We sincerely appreciate your time and effort in reviewing our work. In our previous rebuttal, we have made every effort to appropriately address your concerns. Specifically, based on your valuable feedback, we conducted the suggested experiments to analyze the impact of $f_t$ on the student’s accuracy. These results have been incorporated into the main paper (see Section 4.4, [Add_experiments_impact_of_ft](https://i.imgur.com/QhBf9lh.png))
> > > >
> > > > As the deadline for submitting a revised manuscript approaches, we kindly request your feedback on our rebuttal. We are happy to address any further questions or concerns during the remaining discussion period.

---

> > > > > ### Comment · Reviewer_jiaU · 2024-12-03
> > > > >
> > > > > Thanks for the responses and for conducting new experiments with respect to $f_t$. Through the new results and the original experiments in the manuscript, the authors conclude that a teacher model weak in architecture but strong in training data is the most suitable one. The authors have validated the latter but fail to provide insights for the former, which makes this conclusion quite confusing to me.
> > > > >
> > > > > Also, it seems that similar conclusions have been explored in recent works like [a].
> > > > >
> > > > > [a] A Label is Worth A Thousand Images in Dataset Distillation, Qin et al., NeurIPS 2024.

---

> ### Author Response · Authors · 2024-12-03
>
> We sincerely thank the reviewer for the response and greatly appreciate the clarification provided in the comment.
>
> &nbsp;
>
>
> We would like to emphasize that the theorem and proof presented in our paper pertain specifically to the scenario where both the teacher and the student share identical architectures. Addressing the case where both the teacher's architecture capacity and the size of its dataset are jointly reduced would indeed require further exploration and additional theoretical insights. As this goes beyond the scope of our current work, we will make this distinction clearer in the final version of the paper.
>
>
> &nbsp;
>
> Our paper introduces, for the first time, several intriguing and practical observations about dataset pruning with knowledge distillation. If the reviewer finds these insights and the supporting experiments valuable for the community, we kindly request you to consider revising your score.
>
> Thanks again! :)

---

> > ### Author Response · Authors · 2024-12-03
> >
> > Additionally, we wish to emphasize that while the recent work in [a] is somewhat relevant to ours, their work focuses on dataset distillation, which is a fundamentally distinct field of study from ours (dataset pruning). In their work they show that one simple way to achieve dataset distillation is by randomly selecting a subset of real (i.e., not synthetic) samples, and then soft-labeling them. While this baseline is somewhat similar to our random pruning + KD configuration, they discuss and compare this baseline strictly in the context of dataset distillation. On the other hand, our work focuses on the dynamics of data pruning with KD, and provides a variety of interesting observations in this context. As part of our exploration, one of the things we highlight is the effectiveness of utilizing KD with random pruning by comparing it to a variety of sophisticated data pruning methods. Additionally, we go even further and provide a theoretical motivation for our intriguing observation that employing self-distillation can improve training on pruned data. However, we thank the reviewer for their keen observation, and promise to better address these distinctions in the final version of the paper.
> >
> >
> > **Reference**
> >
> > [a] A Label is Worth A Thousand Images in Dataset Distillation, Qin et al., NeurIPS 2024.

---

### Official Review · Reviewer_5t8M · 2024-11-02

**Soundness:** 2
**Presentation:** 2
**Contribution:** 2
**Rating:** 5
**Confidence:** 4

**Summary:**

Accompanied by a theoretical motivation, the major goal of the paper is to incorporate knowledge distillation to boost the model trained on the pruned dataset. With experiments conducted in image classification, the authors make some observations regarding, e.g., the connection between the pruning factor and the KD weight.

**Strengths:**

1.	The paper is well-organized and easy to follow.
2.	Some empirical observations are intriguing, e.g., distilling with pruned data will exacerbate the gap problem [1], which may provide some insights for further research regarding data pruning and knowledge distillation.
3.	[1] Improved Knowledge Distillation via Teacher Assistant, 2019, AAAI.

**Weaknesses:**

1.It seems the paper is a simple combination of two well-established techniques, and KD has been successfully utilized to boost the model performance. In this way, the overall novelty and contribution are limited. Do the authors have deeper insights regarding the interplay between KD and data pruning, e.g., pruning certain data leads to a better distillation performance (such as [1]), or if can KD be leveraged to identify important samples in data pruning.
[1] Teach Less, Learn More: On the Undistillable Classes in Knowledge Distillation, 2022, NeurIPS.
2.The evaluated KD and data pruning methods are rather outdated. It's necessary to include the latest methods, e.g., [2][3][4].
[2] Decoupled Knowledge Distillation, 2022, CVPR.
[3] Knowledge Distillation from A Stronger Teacher, 2022, NeurIPS.
[4] Data Pruning via Moving-one-Sample-Out, 2023, NeurIPS.

**Questions:**

See weakness.

**Details Of Ethics Concerns:**

No.

---

> ### Author Response · Authors · 2024-11-18
> **Regarding paper’s novelty**
>
> We thank the reviewer for the valuable comments and suggestions.
>
> While knowledge distillation (KD) is a well-established technique commonly used to enhance model performance, to the best of our knowledge, this work is the first to explore the application of KD in the context of dataset pruning, where the teacher is trained on the entire dataset while the student is trained on a smaller pruned subset. We highlight the paper's scenario in the Figure attached in the link below.
>
> As demonstrated in the paper, this scenario reveals several novel and valuable insights. For instance, we observe that simple random pruning outperforms more sophisticated pruning algorithms when KD is applied. Additionally, as noted by the reviewer, we highlight how distilling knowledge to a student trained on pruned data exacerbates the capacity gap problem. These findings provide new perspectives on the interplay between KD and data pruning.
>
> [Figure_sheme](https://i.imgur.com/WcUv4Xw.png)

---

> ### Author Response · Authors · 2024-11-18
> **Regarding additional works**
>
> We find the reviewer’s references highly relevant. The work in [1] is particularly related to the capacity gap problem, and we will include it in our paper. Additionally, we will cite the works in [2, 3, 4] as recommended to further enrich our discussion.
>
>
> **References**
>
> [1] Teach Less, Learn More: On the Undistillable Classes in Knowledge Distillation, NeurIPS 2022.
>
> [2] Decoupled Knowledge Distillation, CVPR 2022.
>
> [3] Knowledge Distillation from a Stronger Teacher, NeurIPS 2022.
>
> [4] Data Pruning via Moving-one-Sample-Out, NeurIPS 2023.

---

> ### Author Response · Authors · 2024-11-21
>
> In response to the reviewer’s comments, we have made several additions to the paper. A revised version has been uploaded, with the new text highlighted in blue for clarity.
>
> Specifically, we have included additional experiments in the Appendix (F), which provide insights into the impact of easy, moderate, and hard pruning on the KD process and the student’s performance in the context of data pruning. Additionally, we have included citations to the recent papers mentioned by the reviewer.

---

> > ### Author Response · Authors · 2024-11-25
> >
> > Dear Reviewer, thank you once again for your time and valuable feedback.
> >
> > We hope our responses have addressed your concerns. Based on your suggestions, we have revised the paper and uploaded the updated version, with the new additions highlighted in blue.
> >
> >
> > We would greatly appreciate it if you could revisit your evaluation and consider raising your score.

---

> > > ### Comment · Reviewer_5t8M · 2024-11-25
> > >
> > > Thanks for the response. After going through the other reviewers’ comments and corresponding responses, I decide to keep my original recommendation of 5. I give credit to some intriguing observations in this paper. However, the absence of deeper insights regarding the interplay between KD and data pruning weakens the overall novelty and contributions. Besides, the suggested experiments are not provided.

---

> > > > ### Author Response · Authors · 2024-11-29
> > > >
> > > > Thank you again for your thoughtful feedback; we highly appreciate your effort and time.
> > > >
> > > >
> > > > Please note that in response to your comment regarding prior works on KD [1, 2], we have added experiments analyzing easy, moderate, and hard pruning with and without KD, across various fraction ratios. These results are detailed in a new section in the Appendix, with a snapshot provided in the link below:
> > > >
> > > > [Add_experiments_easy_moderate_hard_pruning](https://i.imgur.com/4gGKrU5.png)
> > > >
> > > > Specifically, [1] addresses undistillable classes, and [2] explores the role of “difficulty” in KD (TCKD). Accordingly, we have included an experimental section that investigates pruning at different levels of difficulty.
> > > >
> > > > &nbsp;
> > > >
> > > >
> > > > We also believe that our novel observations—such as the finding that smaller teachers can outperform larger ones, and the counterintuitive result that simple random pruning is superior when paired with KD—offer insights that open up new directions in understanding the interplay between KD and data pruning. Further exploration of advanced KD approaches is left as future work.
> > > >
> > > > &nbsp;
> > > >
> > > >
> > > > We would be delighted to provide additional clarifications if needed.
> > > >
> > > > Thank you again for your time and consideration.
> > > >
> > > >
> > > > &nbsp;
> > > >
> > > >
> > > > **Reference**
> > > >
> > > > [1] Teach Less, Learn More: On the Undistillable Classes in Knowledge Distillation, NeurIPS 2022.
> > > >
> > > > [2] Decoupled Knowledge Distillation, CVPR 2022.

---

### Official Review · Reviewer_wDnx · 2024-11-03

**Soundness:** 3
**Presentation:** 3
**Contribution:** 2
**Rating:** 6
**Confidence:** 3

**Summary:**

This paper presents an in-depth investigation into the use of knowledge distillation (KD) for training models on pruned datasets. It provides a comprehensive analysis of the performance of models trained using various dataset pruning strategies and pruning factors across multiple datasets, both with and without the application of KD from their pretrained teachers. Based on the experimental findings, the authors demonstrate that employing a teacher model trained on the full dataset can effectively enhance the performance of student models trained on pruned datasets.

**Strengths:**

1. The experiments in this are comprehensive and sufficient.

2. The theoretical motivation is well-written and reasonable.

3. The paper is easy to read and follow.

**Weaknesses:**

1. The paper has a reference formatting issue. According to the ICLR submission guidelines, `\citep{}` should be used instead of `\cite{}`.

2. The motivation is confusing and needs more clarification. Specifically, this study proposes training a model on a pruned dataset using knowledge distillation from the same model that has already been trained on the full dataset. However, if a well-performing model has already been obtained through training on the complete dataset, it raises the question of __why it is necessary to train the same model from scratch on a pruned dataset__. Providing a stronger rationale for this approach would enhance the clarity and relevance of the study.

3. Although this paper has provided a large volume of experimental results, there lack of insightful analysis. For instance, how different dataset pruning approaches behave differently under the same self-distillation schema can be explained.

4. In Figures 4 and 5, only hard dataset pruning approaches have been compared, while the easy (such as herding) and moderate (such as MoDS) pruning methods are not compared.

5. In Section 4.2, the experiments on adaptive KD weights and pruning factors are only conducted on the CIFAR-100 dataset. The results indicate varying optimal KD weight choices for different pruning factors: a smaller KD weight is preferred when the pruning factor is close to 1, whereas a larger KD weight is favoured when the factor is around 0.1. However, there does not appear to be a consistent pattern underlying this behaviour, making it less practical in real-world datasets. Furthermore, the accuracy gap between different KD weights can exceed 8% from 0.5 to 1.0. It is reasonable to anticipate this accuracy gap escalates on a larger dataset, such as ImageNet-1k. In this case, it is unclear how to choose the optimal weight for a large dataset.

**Questions:**

1. According to W2 and W3, the authors should provide more concrete motivation on why this study is contributive and in what real-world scenarios this analysis can be applied.

2. According to W4 and W5, the authors should provide more experiments to demonstrate the generalizability and practicality of the proposed self-distillation approach.

I'm willing to raise my rating to positive if the authors can provide convincing motivations and necessary experimental results.

---

> ### Author Response · Authors · 2024-11-13
> **Response regarding the motivation**
>
> We thank the reviewer for the constructive feedback. We will try to address the reviewer's concerns.
>
> 1. Regarding your question about the motivation of using KD with a teacher that was already trained using the entire data: "why it is necessary to train the same model from scratch on a pruned dataset?". First, please note that this question may relate to the topic of data pruning in general since typical algorithms for data pruning first train a model (and sometimes even multiple models) on the entire data for obtaining scores for the samples and prune the lowest scores accordingly. However, the question is certainly valid and we will make sure to clarify it in the paper. Specifically, there are some cases where one wishes to re-train a model but cannot use the entire data for some reasons. One practical example (that we have mentioned in the paper in section 3.1, line 252-255) is that the entire dataset is not available anymore, for example due to privacy issues, a large percentage of the data may completely removed from the dataset, while a few more samples are added. Thus, we would like to re-train on the few samples while preserving the knowledge captured by the teacher that was trained on the entire (previous) dataset. This example is related to continual learning. Note that in the paper we show in the first time that using simple random pruning, we can achieve superior performance for all the pruning factors when incorporating KD in the loss. This suggests a highly practical application for cases where portions of the data are gradually removed over time. Another example is HPO - one would like to run with a large number of experiments with different hyper-parameters. Instead of running on the entire data which may require a large computational effort, one can run with only a small subset of the data (e.g. 10%) while using not only the original labels but also the pseudo-labels obtained by the teacher. We will be happy to hear your opinion on these two examples.
>
> 2.  Regarding experiments with adaptive KD weights, please note that our experiments were not limited to the CIFAR-100 dataset. In the appendix, we have provided additional experiments on other datasets. Specifically, we observed and presented a similar trend on both CIFAR-100 and SVHN, demonstrating how setting the KD weight affects accuracy.

---

> > ### Comment · Reviewer_wDnx · 2024-11-16
> >
> > Thank the authors for the response.
> >
> > However, I still have some doubts regarding the scenario described. For instance, if a large portion of data is removed from the dataset while some new data is added, then:
> >
> > 1. The new dataset is no longer a subset of the original dataset, so the conditions outlined in the paper may not apply.
> >
> > 2. While the pre-trained model can still be used, why can't we directly fine-tune the pre-trained model on the new dataset? Training a new model with the same architecture from scratch on the new dataset seems inefficient and time-consuming.
> >
> > The other example of hyperparameter tuning is interesting and seems useful. However, I am curious about how the authors could prove, or potentially explain, that a model's performance using a specific set of hyperparameters on a small portion of the dataset aligns with its performance on the full dataset. In other words, how can we ensure that a model achieving better performance with hyperparameter set A on a subset will also achieve better performance with the same hyperparameters on the entire dataset?
> >
> > In fact, based on my past empirical results of training ViT models on ImageNet-1K, the searched hyperparameters that yield the best performance on a subset of the dataset are often not the optimal ones for training the same model on the full dataset.
> >
> > I'm hoping to hear from the authors.

---

> > > ### Author Response · Authors · 2024-11-18
> > >
> > > Thanks for your response.
> > >
> > > Regarding the first scenario, we agree that the problem relates to continual learning, as mentioned. However, some algorithms and insights from data pruning may also be applicable to continual learning (as also noted in [1, 2]).
> > > Regarding (2), one challenge in directly fine-tuning a pre-trained model on a new dataset is balancing its ability to adapt to new data while retaining prior knowledge. Fine-tuning alone may struggle to effectively update for new samples without risking overfitting or forgetting. In contrast, leveraging pseudo-labels from a teacher model can preserve the knowledge captured from the entire dataset while enabling updates to accommodate new samples.
> > >
> > > Following our discussion on **hyper-parameter selection/optimization (HPO)**, we agree that HPO is indeed one of the significant applications where data pruning plays a crucial role [1].
> > > We agree that the optimal hyper-parameters for a small subset are not necessarily the same as those for the full dataset. However, based on our experience with large datasets, a practical approach could involve performing a coarse search for hyper-parameters on a small data subset (dramatically reducing the search space) followed by a fine-grained search on the full dataset to identify the final optimal hyper-parameters (this process can also be iterative: starting with a very coarse hyper-parameter search using a small data subset, refining the search space based on these results, and then repeating the process with progressively larger subsets until the full dataset is used).
> > >
> > > Another related application of dataset pruning is **neural architecture search (NAS)** (see for example in [1, 3, 4, 5]). These works aim at reducing the search time by performing training on a small subset of the data in each step through the bi-level optimization (as used for example in DARTS).
> > >
> > > If we may add, **active learning** is another highly relevant application where data pruning can make a significant contribution (e.g., [6, 7, 8]).
> > >
> > >
> > >
> > > **References**:
> > >
> > > [1] Shuo Yang et al. Dataset Pruning: Reducing Training Data by Examining Generalization Influence, ICLR 2023
> > >
> > > [2] Yihua Zhang et al. Selectivity Drives Productivity: Efficient Dataset Pruning for Enhanced Transfer Learning, NeurIPS 2023
> > >
> > > [3] Xiyang Dai et al. DA-NAS: Data Adapted Pruning for Efficient Neural Architecture Search, 2020
> > >
> > > [4] Chongjun Tu et al. Efficient Architecture Search via Bi-level Data Pruning, 2023
> > >
> > > [5] Vishak Prasad C et al. Speeding up NAS with Adaptive Subset Selection, 2022
> > >
> > > [6] Ravi S Raju, Accelerating Deep Learning with Dynamic Data Pruning, 2021
> > >
> > > [7] Abdul Hameed Azeemi, Language Model-Driven Data Pruning Enables Efficient Active Learning, 2024
> > >
> > > [8] Yichen Xie et al. Active Finetuning: Exploiting Annotation Budget in the Pretraining-Finetuning Paradigm, 2023

---

> > > > ### Comment · Reviewer_wDnx · 2024-11-19
> > > >
> > > > I thank the authors for the prompt response, which partially addresses my doubts concerning the significance of this paper. I'm willing to raise my score to positive.

---

> > > > > ### Author Response · Authors · 2024-11-21
> > > > >
> > > > > Thank you for your response to our rebuttal and for raising your score.
> > > > >
> > > > > Following your suggestion, we have further added experiments in the Appendix (F) comparing easy, moderate, and hard pruning, along with the corresponding insights. Additionally, we have addressed the reference formatting issue—thank you for bringing it to our attention.
> > > > >
> > > > > Please note that we have uploaded a revised version of the paper, with the added text highlighted in blue.

---

### Meta-Review · Area_Chair_WFfc · 2024-12-21

**Metareview:**

This paper investigates the use of knowledge distillation (KD) to improve the training of neural networks on pruned datasets. It demonstrates that incorporating KD from a teacher model trained on the full dataset can enhance model performance across various datasets, pruning methods, and pruning fractions. The study also establishes a relationship between the pruning factor and the optimal KD weight, offering practical insights, such as the potential benefits of using smaller teacher models in lower pruning regimes. Experiments on datasets like CIFAR, SVHN, and ImageNet support these findings.

The strengths of this paper include its clear and well-organized writing, recognized by all reviewers, as well as its comprehensive experimental evaluation and practical insights. The main weakness of this paper is its limited novelty, as it primarily combines existing methods without sufficient theoretical validation.

This paper received borderline reviews leaning towards negative (one 6 with a confidence of 3, and two 5s with a confidence of 4). During the AC-reviewer discussion after the rebuttal, Reviewer wDnx adopted a neutral stance, Reviewer jiaU maintained a negative position, and Reviewer 5t8M did not respond to the discussion and kept the original score of 5. Therefore, this paper is decided to be rejected.

**Additional Comments On Reviewer Discussion:**

Reviewer wDnx asked for more motivation and additional experimental results. The authors provided explanations and references, leading to wDnx increasing their score from negative to positive. However, the other two reviewers did not change their scores during the rebuttal, citing the lack of novelty as the main issue.

---

### Decision · Program_Chairs · 2025-01-22

Reject